# Canonical Transient Receptor Potential Channel 3 Contributes to Cerebral Blood Flow Changes Associated with Cortical Spreading Depression in Mice

**DOI:** 10.3390/ijms241612611

**Published:** 2023-08-09

**Authors:** Fang Zheng

**Affiliations:** Department of Pharmacology and Toxicology, College of Medicine, University of Arkansas for Medical Sciences, Little Rock, AR 72205, USA; zhengfang@uams.edu

**Keywords:** TRPC3, spreading depression, high potassium, NMDA, needle prick, laser-speckle contrast imaging

## Abstract

Cortical spreading depression is a pathophysiological event shared in migraines, strokes, traumatic brain injuries, and epilepsy. It is associated with complex hemodynamic responses, which, in turn, contribute to neurological problems. In this study, we investigated the role of canonical transient receptor potential channel 3 (TRPC3) in the hemodynamic responses elicited by cortical spreading depression. Cerebral blood flow was monitored using laser speckle contrast imaging, and cortical spreading depression was triggered using three well-established experimental approaches in mice. A comparison of TRPC3 knockout mice to controls revealed that the genetic ablation of TRPC3 expression significantly altered the hemodynamic responses elicited using cortical spreading depression and promoted hyperemia consistently. Our results indicate that TRPC3 contributes to hemodynamic responses associated with cortical spreading depression and could be a novel therapeutic target for a host of neurological disorders.

## 1. Introduction

Cortical spreading depression is a common phenomenon in many neurological disorders [1,2,3,4]. It has been proposed to be the underlying mechanism for migraine [5] and is thought to be involved in the pathophysiology of seizure [6], stroke [7], and traumatic brain injury [4]. Therefore, a detailed mechanistic understanding of cortical spreading depression and associated hemodynamic responses is important for developing better treatment options for a host of neurological disorders.

Cortical spreading depression involves the massive depolarization of neurons and glial cells accompanied by changes in local cerebral blood flow (CBF) [8,9]. The hemodynamic response to spreading depression is complex, and at least three vasomotor components have been identified: (a) a vasoconstrictive response that overlaps with spreading depression temporally; (b) a profound hyperemia (i.e., an increase in CBF) that starts at or soon after the depolarization onset; and (c) a post-depression oligemia (i.e., a decrease in CBF) that could last up to an hour [9,10]. Each hemodynamic response is an aggregate of responses originating from one or more components of the neurovascular unit [9,10].

The mediators and permissive modulators of the hemodynamic response to cortical spreading depression have been subject to much investigation. Modulators that augment or attenuate each of the three vasomotor components have been well documented [10]. However, the primary mediator of each component remains elusive.

The canonical transient receptor potential channel (TRPC) family is the mammalian homolog of the drosophila *trp* gene [11,12,13]. Mammalian TRPC genes encode non-selective, calcium-permeable cation channels that are widely expressed in both central nervous systems and peripheral tissues [14,15]. Among the seven family members of TRPC, TRPC3 is the most abundant TRPC channel in the brain [16] and is expressed in neurons [17,18,19,20], astrocytes, and other cell types in the cerebral vasculature [21]. We have reported previously that TRPC3 channels in smooth muscle cells in brain vasculature contribute to neurovascular coupling dysfunction caused by status epilepticus [22]. To fully evaluate the potential of TRPC3 as a novel therapeutic target for neurovascular decoupling, its role in other neurological disease models need to be explored. Given the fact that it is a shared pathophysiological phenomenon in seizure, stroke, and traumatic brain injury, cortical spreading depression represents a high-priority model to investigate the potential contribution of TRPC3 channels.

To determine the contribution of TRPC3 channels to hemodynamic responses associated with cortical spreading depression, we compared TRPC3 global knockout mice to pooled control mice, which include wild-type mice and TRPC3flx^+/+^ littermate controls. Cortical spreading depression was induced chemically by high K^+^, or high concentrations of NMDA as described previously [8]. Furthermore, cortical spreading depression was also induced by needle pricking, a mechanical injury [23]. Our data showed that the genetic ablation of TRPC3 channel expression significantly changed hemodynamic responses to cortical spreading depression regardless of the experimental approach used to induce it. Our results indicate that TRPC3 channels are significant contributors to neurovascular coupling dysfunction associated with cortical spreading depression.

## 2. Results

To investigate the role of TRPC3 channels in hemodynamic responses associated with spreading depression, a cranial window was surgically created to provide direct access to drugs, and cerebral blood flow (CBF) was monitored with laser-speckle contrast imaging in anesthetized mice. We used high K^+^ as the primary experimental approach to induce cortical spreading depression experimentally. KCl solution (2M) was diluted in saline to make various stock solutions with different low K^+^ concentrations, and the stock solution was applied as a bolus to the cranial window to elicit responses. The estimated peak concentration of K^+^ was 1/10 of the stock concentration. As shown in Figure 1, low concentrations of K^+^ elicited an increase in CBF, whereas high concentrations of K^+^ elicited an initial transient decrease in CBF, followed by a transient increase in CBF. The observed CBF changes to high K^+^ in littermate control mice were generally consistent with the three distinct components proposed by Ayata [9].

The CBF changes elicited by low K^+^ in TRPC3 KO were similar to what was observed in control mice (Figure 2). The amplitudes of CBF increases elicited by 2 mM K^+^ or 6 mM K^+^ in TRPC3KO and control mice were comparable (Figure 2a,b). The transient decreases in CBF elicited by high K^+^ (50 mM) in TRPC3KO and control mice were also similar (Figure 2c). However, a prolonged hyperemia after the initial CBF decrease was observed in TRPC3KO mice.

The sampling rate (1 LSI image/4 s) used in our initial experiments (Figure 1 and Figure 2) did not provide enough temporal resolution for a detailed analysis of the complex CBF changes elicited by high K^+^. Therefore, we monitored CBF using a higher sampling rate (1 LSI image/s) and compared the hemodynamic responses to cortical spreading depression elicited by high K^+^ in TRPC3KO mice and control mice (Figure 3). Pooled data from TRPC3 KO mice and controls revealed highly significant differences between the two groups at two distinct phases: (a) the early phase of the vasoconstriction and (b) hyperemia after the initial vasoconstriction. It appeared that the onset of vasoconstriction was significantly delayed in TRPC3KO mice, and the hyperemia was augmented in TRPC3KO mice.

The faster temporal resolution permitted a more accurate determination of the peak decrease and peak increase in CBF elicited by high K^+^. Figure 4 shows a quantitative comparison between the TRPC3KO mice and controls. Although the initial vasoconstriction was slightly reduced, there was no statistically significant difference between TRPC3KO mice and controls (Figure 4a). The peak of hyperemia showed a trend of increase in TRPC3KO mice (Figure 4b). However, the difference between TRPC3KO and WT was not significant due to high variability in both groups. The most prominent difference between TRPC3KO mice and control mice was the prolonged hyperemia in TRPC3KO, which lasted for at least 4 min (Figure 4c). Collectively, our results clearly showed that genetic ablation of TRPC3 channel expression significantly altered the hemodynamic responses associated with high K^+^-induced cortical spreading depression, and the main effect was an augmentation and prolongation of hyperemia.

To determine how widely applicable our findings regarding the role of TRPC3 channels in spreading depression-induced hemodynamic responses are, we sought to expand our study to include additional experimental models of spreading depression. NMDA, a selective agonist for the NMDA subtype of ionotropic glutamate receptor, was another chemical agent that was used to elicit cortical spreading depression previously [5]. We tested various concentrations of NMDA with our cranial window preparation and monitored the CBF changes using laser-speckle contrast imaging. In control mice, NMDA always elicits a transient increase in CBF, without the initial transient decrease consistently observed after high K^+^ treatment (Figure 5a). In TRPC3KO mice, a greater and more sustained increase in CBF was observed (Figure 5b). These observations suggest that TRPC3 channels are also involved in the hemodynamic responses associated with high NMDA-induced cortical spreading depression.

Another frequently utilized animal model of cortical spreading depression is the needle prick model. In control mice, needle prick elicited a developing hyperemia that reaches its peak approximately in 2–3 min (Figure 6a). In TRPC3KO mice, the opposite typically occurred, a prominent hyperemia emerged after the needle prick (Figure 6a). This difference between TRPC3KO mice and control mice was confirmed via a quantitative comparison of the CBF changes 2–3 min after the needle prick (Figure 6b). The genetic ablation of TRPC3 channel expression converted a consistent decrease in CBF into significant increases in CBF.

In summary, we investigated the role of TRPC3 channels in the hemodynamic responses associated with cortical spreading depression using three well-established mouse models of spreading depression. Significant differences between TRPC3KO mice and control mice are seen in all three models.

## 3. Discussion

In this study, we investigated the potential role of TRPC3 channels in hemodynamic responses to induced cortical spreading depression using three different experimental approaches. As well documented in the literature, the hemodynamic responses to cortical spreading depression in these three experimental approaches exhibit distinct characteristics. The CBF changes associated with cortical spreading depression evoked by high K^+^ are the most complex, with possibly at least three distinct components. The CBF changes associated with cortical spreading depression evoked by high NMDA are primarily hyperemia, without even a hint of the initial vasoconstriction and decrease in CBF consistently seen in high K^+^-treated mice. The CBF changes associated with cortical spreading depression elicited with a needle prick are dominated by a decrease that lasts several minutes. Although the hemodynamic responses to cortical spreading depression elicited using different experimental approaches are distinct, the genetic ablation of TRPC3 channel expression significantly altered the observed CBF changes regardless of the experimental approaches used. This remarkable consistency strongly suggests that TRPC3 channels are an important contributor to hemodynamic responses to cortical spreading depression.

What are the exact roles of TRPC3 channels in hemodynamic responses to cortical spreading depression? The augmentation of hyperemia has emerged as a common event in CBF changes associated with cortical spreading depression elicited using three different experimental approaches. Ayata has proposed three distinct vasomotor responses to high K^+^-induced cortical spreading depression, i.e., an initial vasoconstriction, followed by a transient vasodilation and an oligemia [9]. Our data showed that the initial decrease in CBF elicited by high K^+^ was not significantly reduced (Figure 4a), although it may be delayed (Figure 3). This is somewhat expected because the vasoconstriction responsible for this decrease in CBF is likely mediated via the depolarization of smooth muscle cells in cerebral vasculature by high K^+^ directly. Although the hyperemia associated with high K^+^-induced cortical depression appeared greater in TRPC3 KO mice, the observed difference was not statistically significant (Figure 4b). On the other hand, hyperemia lasted significantly longer in TRPC3 KO mice (Figure 4c). The most plausible interpretation of our results is that TRPC3 channels are a major contributor to the proposed “oligemia” proposed by Ayata. The genetic ablation of TRPC3 channel expression would augment and prolong hyperemia by ameliorating oligemia associated with cortical spreading depression. This hypothesis would also provide an explanation for the effects of TRPC3 knockout on CBF changes elicited via cortical spreading depression associated with high NMDA treatment. In that case, the onset of the oligemia mediated via TRPC3 channels would start to slowly decrease the observed hyperemia and make it appear transient. The genetic ablation of TRPC3 expression would abolish oligemia and prolong hyperemia (Figure 5). Although the hypothesis that TRPC3 channels play a critical role in oligemia associated with cortical spreading depression is consistent with our results from two chemical cortical spreading models, it is problematic with mechanical injury models such as the needle prick model of cortical spreading depression. The hemodynamic response to needle prick in mice lasted only a few minutes and CBF returned to baseline afterward. Therefore, oligemia is not observed after needle pricking. The absence of oligemia did not prevent genetic ablation of TRPC3 from uncovering hyperemia after the needle injury (Figure 6), and our results from the needle prick model suggest that TRPC3 channels are involved in additional vasomotor responses. Future studies are needed to reveal the nature and role of such vasomotor responses.

TRPC3 channels are highly expressed in all major cell types in the neurovascular unit, in particular, mural cells (smooth muscle cells and pericytes) in cerebral vasculature [21,24]. Native TRPC3 channels in the brain are predominantly homomeric tetramers [16]. The activation of TRPC3 channels in mural cells could be a possible mechanism for cortical-spreading-depression-induced oligemia. Since TRPC3 channels are non-selective cation channels, the activation of TRPC3 will lead to membrane depolarization and the activation of voltage-gated calcium channels to promote mural cell contraction. TRPC3 channels are also highly calcium permeable; therefore, the activation of TRPC3 could also directly contribute to an increase in intracellular free calcium concentration that augments mural cell contraction. Although this mural cell hypothesis is attractive and plausible, there are alternative hypotheses that could explain the observed role of TRPC3 in hemodynamic responses associated with cortical spreading depression. Astrocytes express high levels of angiotensin [21], and TRPC3 channels expressed in astrocytes may be involved in the synthesis and release of angiotensin. Endothelial cells also express TRPC3 channels [21], and these channels may be involved in the synthesis and release of endothelin 1. These two vasoconstrictors could be mediators of vasoconstriction associated with cortical spreading depression. Future studies with a cell-type specific knockout of TRPC3 will allow us to pinpoint the exact cell type that can account for the augmented hyperemia observed in TRPC3 global KO mice.

There are multiple possible signaling pathways that could lead to the activation of TRPC3 channels by cortical spreading depression. The massive depolarization of neurons and astrocytes during cortical spreading depression causes the release of many vasoactive neurotransmitters and peptides [10]. In addition to angiotensin and endothelin 1, NPY is another strong vasoconstrictor for cerebral arteries and arterioles [25,26]. It can be released from interneurons, sympathetic nerve endings, and astrocytes [26,27,28]. Mural cells exhibit high levels of NPY R1 receptor expression [29]. Therefore, NPY is a good candidate for the upstream signaling molecule of TRPC3 channels in mural cells. Clearly, significant amounts of future work are needed to identify the upstream signaling pathways that are responsible for the observed TRPC3′s role in hemodynamic responses associated with the cortical spreading depression.

Although many questions remain to be answered, our results revealed an intriguing role of the TRPC3 channel in hemodynamic responses associated with cortical spreading depression. Our findings in this study and previously published results on TRPC3′s role in seizure-induced inverse hemodynamic response [22] suggest that the TRPC3 channel is potentially a novel therapeutic target for the treatment of neurovascular coupling dysfunction. The development of small molecule modulators of TRPC channels has progressed rapidly in recent years [30,31,32]. Hopefully, a selective TRPC3 channel inhibitor with good pharmacokinetic properties will emerge and can be tested in clinical trials.

## 4. Materials and Methods

***Animals***: Adult male mice (3–4 months old) were used in all experiments. Wild-type mice, TRPC3flx^+/+^ mice, and TRPC3 knockout mice were all in a mixed genetic background (C57Blk/129S).

***Cranial Window Preparation and Laser Speckle Contrast Imaging (LSI)***: Mice were initially anesthetized with sevoflurane (5%) and then maintained by urethane (1.1 mg/kg, i.p.). Anesthetized mice were affixed to a stereotaxic frame, and a cranial window was surgically created for laser speckle contrast imaging and drug application to the cortex. Briefly, the head was shaved, and the shaved skin was then wiped with 70% alcohol pads. The skin over the skull was removed using surgical incisions (1/2 inch), and the soft tissues over the skull bone were scrapped off to expose the bone surface. A high-speed drill with 0.7 mm carbon steel burrs was used to make an oval groove (2–3 mm in diameter) in the skull over the somatosensory cortex, and the skull bone in the center was removed to expose the brain. The dura exposed by the cranial window was removed using sharp forceps to allow direct drug access to the brain tissues.

Cerebral blood flow was measured using a laser speckle contrast imaging device (moorFLPI-2, Moor Instruments, Axminster, UK). The sampling rate was either 1 frame for every 4 s or 1 frame for every 1 s. The slow sample rate provided better spatial resolution, whereas the faster sample rate provided better temporal resolution with a tradeoff of spatial resolution. An oval-shaped region of interest (ROI) in the center of the cranial window was selected, and the flux value from the ROI was used to monitor the changes in cerebral blood flow.

***Mouse Model of Chemical-induced Spreading Depression***: To induce spreading depression, a bolus of high KCl stock solution or high NMDA stock solution was applied directly to the cortex exposed by the cranial window, and the final concentration of KCl or NMDA was estimated to be 1/10th of the concentration of the stock solution applied. This dilution factor of 10 was based on the estimated volume ratio between the cranial window and the bolus and empirically confirmed by the observed responses to KCl and NMDA. After 5 min, KCl or NMDA was washed out with Earle’s balanced salt solutions. The cerebral blood flow was monitored using LSI before, during, and after exposure to high K^+^ or NMDA.

***Mouse Model of Mechanical-Injury-induced Spreading Depression***: To induce spreading depression with mechanical injuries, a 25 and ½ gauge needle was inserted into the cortex for approximately 0.5 mm. Minimal bleeding was caused by needle injury for mice included in this study. Animals with excessive bleeding were excluded from data analysis. CBF was monitored with LSI before and after needle prick at 1 frame/s.

## 5. Conclusions

We used three well-established mouse models of cortical spreading depression to investigate the roles of TRPC3 channels in hemodynamic responses. A comparison of the cerebral blood flow changes revealed significant differences between TRPC3KO mice and control mice in all three models. The genetic ablation of TRPC3 channel expression consistently led to hyperemia after cortical spreading depression. Collectively, our results indicate that TRPC3 channels contribute to the pathophysiology of cortical spreading depression.

## Figures and Tables

**Figure 1 ijms-24-12611-f001:**
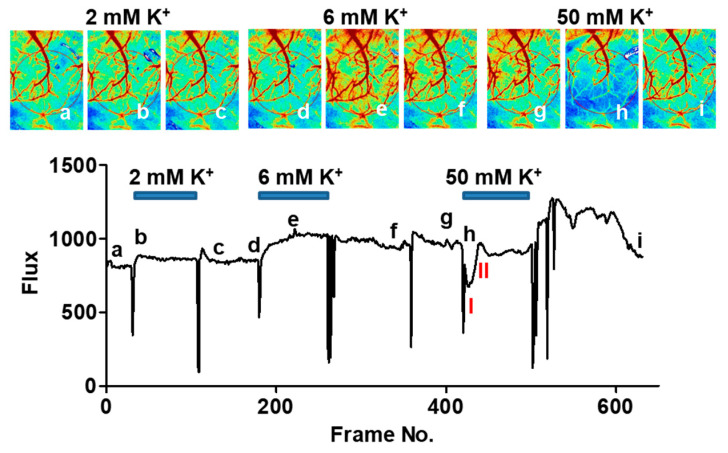
Representative CBF responses in a littermate control mouse to different concentrations of K^+^ applied to a cranial window. An LSI image was collected every 4 s and the flux (CBF measured using LSI) of the region of interest (indicated with the red oval was plotted in the lower panel. The timing of drug applications and washout was indicated with artifacts in the LSI imaging. Images at the top were taken at indicated time points (**a**–**i**) in the bottom panel. Note that lower concentrations of K^+^ results in an increase in CBF whereas high K^+^ results in a transient decrease (I) in CBF followed by a transient increase (II).

**Figure 2 ijms-24-12611-f002:**
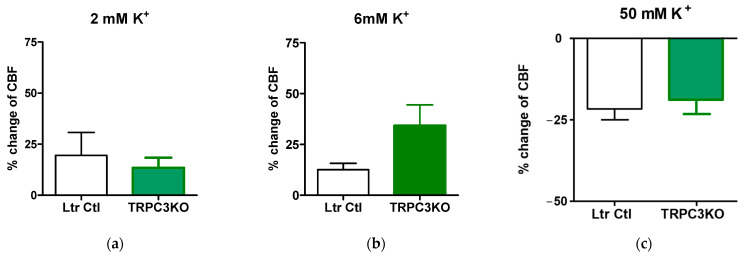
Quantitative comparisons of CBF changes elicited by (**a**) 2 mM KCl; (**b**) 6 mM KCl; and (**c**) 50 mM KCl. There is no statistically significant difference between control mice (*n* = 7) and TRPC3KO mice (*n* = 7).

**Figure 3 ijms-24-12611-f003:**
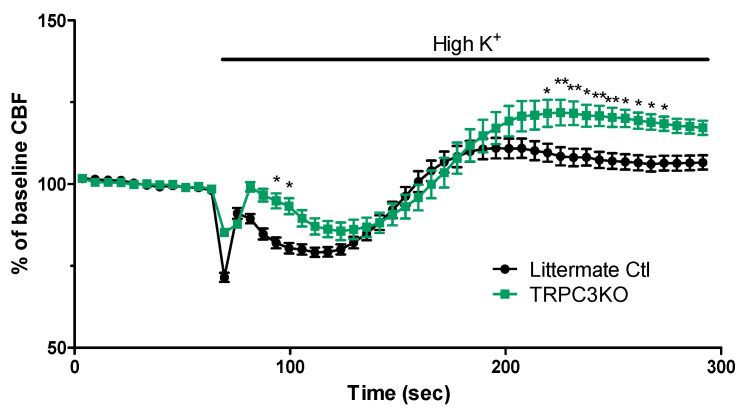
Comparison of CBF changes associated with high K^+^-induced spreading depression in TRPC3KO (*n* = 6) and control mice (*n* = 7). Note that there is a highly significant genotype effect (*p* < 0.001, Two-way ANOVA; *: *p* < 0.05, **: *p* < 0.01, Bonferroni posttests).

**Figure 4 ijms-24-12611-f004:**
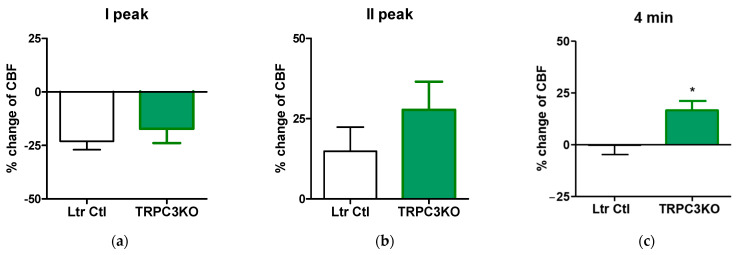
Quantitative comparison of 3 major phases of hemodynamic responses associated with high K^+^-induced cortical depression between TRPC3KO mice (*n* = 6) and controls (*n* = 7). The first (I) and second (II) changes in CBF were quantified (**a**,**b**) and the late CBF changes (4 min after high K^+^ bolus) were also measured (**c**). *: *p* < 0.05, unpaired *t*-test.

**Figure 5 ijms-24-12611-f005:**
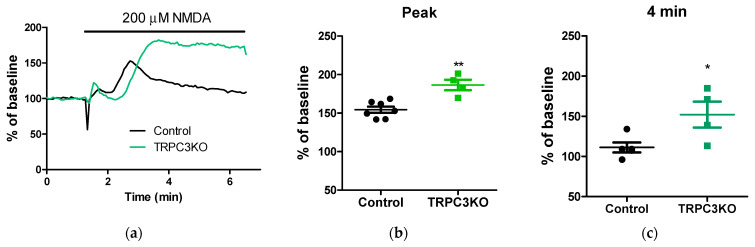
Hemodynamic responses associated with high NMDA-induced cortical depression. (**a**) Representative CBF changes after bolus application of NMDA in TRPC3KO mice and controls. Note that a sustained increase in CBF was observed in TRPC3KO mice, whereas a transient increase in CBF was typical in control mice. (**b**) Comparison of the peak CBF increase in TRPC3KO mice (*n* = 4) and controls (*n* = 7). **: *p* < 0.01, unpaired *t*-test. (**c**) Comparison of the late CBF changes (approximately 4 min after NMDA application) in TRPC3KO mice (*n* = 4) and controls (*n* = 7). *: *p* < 0.05, un-paired *t*-test.

**Figure 6 ijms-24-12611-f006:**
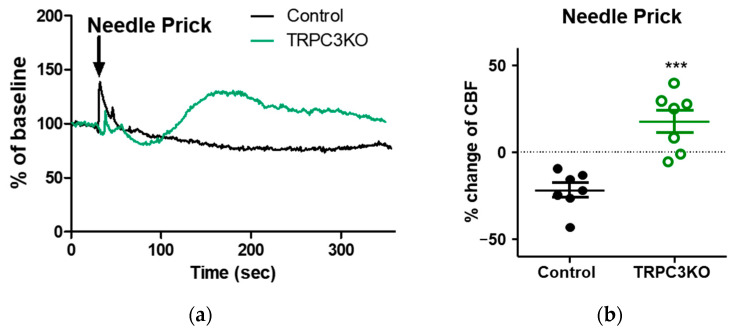
Hemodynamic responses associated with needle-prick-induced cortical depression. (**a**) Representative CBF changes after a needle prick in TRPC3KO mice and controls. (**b**) Comparison of the CBF changes 2–3 min after needle prick in TRPC3KO mice (*n* = 6) and controls (*n* = 7). ***: *p* < 0.001, unpaired *t*-test.

## Data Availability

Data supporting reported results will be available by contacting the author.

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
