# Peer review of "Canonical Transient Receptor Potential Channel 3 Contributes to Cerebral Blood Flow Changes Associated with Cortical Spreading Depression in Mice"

_ijms, 2023, doi:10.3390/ijms241612611_

Round 1
Reviewer 1 Report
If it is the journal’s policy that the description of methodology should come at the end of the full text of the paper, I would have found it easier on first reading if, at the beginning of your Results section, where you gave a few details of methodology, you had indicated that there were further details later in the text. Not knowing this on the initial reading gave me difficulty in trying to interpret some of the results until the paper was re-read.
I also had difficulty with Figure 1, and remain unclear as to whether it applied to the knockout mice only and whether it was intended mainly to illustrate the sort of data obtained rather than to present a result.
You indicate that you were working with a well-established mouse model of cortical spreading depression. I have no experience with these models, but from your description it seems to me that only the needle prick injury technique, the one that Leao first used many years ago, may have initiated as the primary event cortical spreading depression that becomes recognisable by the subsequent associated spreading vascular changes, while your other methods of induction may have primarily affected the vasculature, whether or not the cortex became involved. As far as I can see, your methodology showed progressive vascular changes at one site but did not necessarily show spreading of the changes or alteration of cerebral cortical function, though the latter might be presumed in relation to the needle injury. I raise this because you sometimes seemed to specify that it was the vascular change associated with cortical spreading depression that you are dealing with, but at other times only mention cortical spreading depression. From the point of view of future treatment of human illness many would see the consequences of the cortical depression as the main issue and the associated vascular changes as secondary and of arguable consequence and I wonder whether this should be taken into consideration in speculating where your study might lead in the future.
Author Response
I would like to thank both reviewers for their careful reading of the manuscript and their thoughtful comments. Reviewer 1’s concerns were addressed below.
“If it is the journal’s policy that the description of methodology should come at the end of the full text of the paper, I would have found it easier on first reading if, at the beginning of your Results section, where you gave a few details of methodology, you had indicated that there were further details later in the text. Not knowing this on the initial reading gave me difficulty in trying to interpret some of the results until the paper was re-read.”
It is the Journal’s policy that the Method section came after the results. However, I have modified the text to make it easier to follow the result section by adding a brief description of the general approach at the beginning of the result section.
“I also had difficulty with Figure 1, and remain unclear as to whether it applied to the knockout mice only and whether it was intended mainly to illustrate the sort of data obtained rather than to present a result.”
Figure 1 was intended mainly to illustrate the type of hemodynamic responses to various concentration of K+, and the ROI used to calculate the mean flux value.
“You indicate that you were working with a well-established mouse model of cortical spreading depression. I have no experience with these models, but from your description it seems to me that only the needle prick injury technique, the one that Leao first used many years ago, may have initiated as the primary event cortical spreading depression that becomes recognisable by the subsequent associated spreading vascular changes, while your other methods of induction may have primarily affected the vasculature, whether or not the cortex became involved. As far as I can see, your methodology showed progressive vascular changes at one site but did not necessarily show spreading of the changes or alteration of cerebral cortical function, though the latter might be presumed in relation to the needle injury. I raise this because you sometimes seemed to specify that it was the vascular change associated with cortical spreading depression that you are dealing with, but at other times only mention cortical spreading depression. From the point of view of future treatment of human illness many would see the consequences of the cortical depression as the main issue and the associated vascular changes as secondary and of arguable consequence and I wonder whether this should be taken into consideration in speculating where your study might lead in the future.”
Reviewer 1 is correct that although K+ induces cortical depression, there is a direct effect of high K+ on cerebral vasculature. It is well known in the field that the hemodynamic changes associated with spreading depression show differences when different methods were used. Our data confirmed that. This is exactly the reason why we used three different methods to induce spreading depression in order to assess the potential role of TRPC3 channels. The focus of this study is indeed the hemodynamic responses associated with spreading depression. I have searched and edited the manuscript and made sure that it has been consistently stated.
Reviewer 2 Report
Q1. Fig 1. Please clearly explain the meaning of a to i (a time course, or represent variable treatments ), and the units of flux and x-axis. Does the tracing in the lower side represent the quantification of the upper pictures? Please explain clearly.
Q2. The time gap between 2mM, 6mM, and 50mM is variable. why? Has the author considered the possibility of “desensitisziation”?. You should provide a regular and repetitive treatments time course to prove that there is no desensitization problem, and then your results are reliable.
Q3. You should have a “normalization” treatment in the end of the experiment (Figure 1 and Figure 3)
Q4. Why in your Fig 2 KCL 6mM evoked a stronger response than 2mM, but this effect was not prominent in the Fig 1 (Frame No.)?
Q5. the purpose for the “4 min” of the treatment (Fig 4C)
Author Response
I would like to thank both reviewers for their careful reading of the manuscript and their thoughtful comments. Reviewer 2’s concerns were addressed below.
Q1. Fig 1. Please clearly explain the meaning of a to i (a time course, or represent variable treatments), and the units of flux and x-axis. Does the tracing in the lower side represent the quantification of the upper pictures? Please explain clearly.
The images were chosen to represent the effects of different K+ treatments. Flux value from LSI has no unit and its changes indicate the changes of CBF. The flux value plotted in the lower panel was from the region of interest indicated in the images (red oval). I have edited the legend of Figure 1 to clarify these points.
Q2. The time gap between 2mM, 6mM, and 50mM is variable. why? Has the author considered the possibility of “desensitization”? You should provide a regular and repetitive treatments time course to prove that there is no desensitization problem, and then your results are reliable.
We wait until the CBF returns to baseline after washout before applying K+ again. With higher K+, it appears that that it takes longer time for CBF to return to baseline, thus the longer time gap. Extracellular K+ directly changes the membrane potential thus there is no “desensitization”. We have tested whether there is a desensitization when NMDA was applied repeatedly and found the comparable CBF changes by the same concentration of NMDA. I would also like to emphasize that Figure 1 is presented to illustrate the type of hemodynamic responses to various concentrations of K+. The quantification of high K+ induced hemodynamic responses was done with a different set of data (Figure 3 and 4).
Q3. You should have a “normalization” treatment in the end of the experiment (Figure 1 and Figure 3).
We apply needle prick to induce spreading depression at the end of each experiment.
Q4. Why in your Fig 2 KCL 6mM evoked a stronger response than 2mM, but this effect was not prominent in Fig 1 (Frame No.)?
Figure 1 showed a stronger response to 6 mM K+ (image e) than 2 mM (Image b). Figure 2 showed comparisons between TRPC3KO and controls.
Q5. the purpose for the “4 min” of the treatment (Fig 4C)
The 4 min time point was chosen to capture the longer lasting hemodynamic responses associated with spreading depression.